# Nationwide trends and regional variation in intravitreal injections and anti-VEGF agent use in Japan from 2017 to 2022: An analysis of the NDB Open Data

Yoshiaki Kabata [ID][1]*, Ryo Terauchi[2], Tadashi Nakano[2]

1 Department of Ophthalmology, Jikei University School of Medicine, Daisan Hospital, Tokyo, Japan,
2 Department of Ophthalmology, Jikei University School of Medicine, Tokyo, Japan

* kabata22122@gmail.com

## Abstract

Intravitreal injections are an essential treatment modality for retinal vascular and macular diseases, yet nationwide evidence describing long-term trends, regional variation, drug composition, seasonality, and the impact of the COVID-19 pandemic in Japan remains limited. We conducted a nationwide retrospective observational study using aggregated National Database of Health Insurance Claims and Specific Health Checkups (NDB) Open Data for fiscal years (FYs) 2017–2022 (April–March). Total intravitreal injection procedures were identified using procedure code G016, and drug-specific counts of aflibercept, ranibizumab, brolucizumab, and faricimab were extracted using drug codes. Crude utilization rates (per 100,000 population) were calculated using official population estimates. Annual trends were assessed using linear regression, regional variation was summarized descriptively and compared across predefined geographic groupings, monthly patterns were evaluated using Friedman tests, and the pandemic impact was examined by comparing FY 2019 (pre-COVID), FYs 2020–2021 (COVID period), and FY 2022 (post-COVID). National total intravitreal injection procedures (G016) increased from 600,068 in FY 2017–936,715 in FY 2022 (anti-VEGF injections: 591,469–929,704), corresponding to an average increase of approximately 65,000 injections per year. The mean crude utilization rates rose from 478.6 ± 109.2 to 775.7 ± 143.1 per 100,000 population, while relative inter-prefectural variability decreased modestly (coefficient of variation 0.228 to 0.184). Aflibercept remained the predominant anti-VEGF agent, with diversification after the introduction of brolucizumab (FY 2020) and faricimab (FY 2022). Monthly volumes showed modest seasonality, with higher counts toward the end of the fiscal year. Injection volumes showed no marked decline during the pandemic; April–May 2020 totals (119,100 injections) were only 0.9% lower than the same months in 2019 (120,202 injections). These findings indicate sustained growth in intravitreal injection

**Data availability statement:** All derived datasets and analysis tables generated for this study have been deposited on Figshare (DOI: https://doi.org/10.6084/m9.figshare.31223410). The original source data are publicly available as aggregated tables from Japan's National Database of Health Insurance Claims and Specific Health Checkups (NDB) Open Data and official population statistics; this figshare record contains the processed tables and outputs needed to reproduce the results. All analyses were conducted using these publicly accessible sources, and no individual-level data were used.

**Funding:** The author(s) received no specific funding for this work.

**Competing interests:** The authors have declared that no competing interests exist.

activity in Japan, persistent but modestly narrowing regional variation, and maintenance of injection services during COVID-19.

## Introduction

Retinal vascular and macular diseases such as age-related macular degeneration (AMD), diabetic macular edema, macular edema secondary to retinal vein occlusion, and myopic choroidal neovascularization are major causes of blindness and severe visual impairment worldwide [1–5]. Intravitreal injections of anti–vascular endothelial growth factor (VEGF) agents have become the first-line treatment for these conditions, and numerous clinical trials and real-world studies have demonstrated their efficacy in improving visual outcomes and reducing the risk of blindness [6–9]. In Japan, a rapidly aging population is expected to increase the number of patients with AMD and other retinal diseases [10,11]. Understanding the demand for anti-VEGF therapy and its impact on healthcare resources is therefore critical from both clinical and policy perspectives [10–12].

In Japan, the anti-VEGF agents aflibercept and ranibizumab were the first to become available, followed by brolucizumab in 2020 and faricimab in 2022. These drugs differ in their pharmacokinetics, mechanisms of action, dosing intervals, and approved indications, and these differences may influence treatment strategies and clinical practice patterns. Anti-VEGF therapy usually requires long-term, repeated intravitreal injections, and various regimens including treat-and-extend and pro re nata (PRN) approaches are employed [10,11]. Consequently, the actual implementation of anti-VEGF therapy in a given region or institution reflects not only the number of patients but also treatment algorithms, healthcare delivery systems, and patients' healthcare-seeking behavior.

Intravitreal anti-VEGF injections are mainly performed in the outpatient setting; therefore, a continued increase in the number of injections may pose a substantial burden on outpatient services and healthcare costs [13]. Because a large proportion of the treated population is elderly, regional differences in demographic structure and the distribution of ophthalmic resources may have a significant impact on access to anti-VEGF treatment [13,14]. Several studies from Japan and other countries have examined real-world anti-VEGF use based on single-center or multicenter registries, but their study populations and catchment areas have been limited [3,10–12]. As a result, there are still relatively few reports that describe nationwide temporal trends and regional variation in injection volumes. In addition, large-scale nationwide analyses of drug-specific utilization and inter-prefectural variation in Japan are limited.

In recent years, part of the National Database of Health Insurance Claims and Specific Health Checkups of Japan (NDB) has been released as open data, enabling nationwide evaluation of procedure-specific volumes. Previous studies using NDB Open Data have analyzed temporal trends and the impact of the coronavirus disease 2019 (COVID-19) pandemic on ophthalmic surgeries such as cataract extraction, vitrectomy, and glaucoma surgery [15]. However, nationwide analyses focusing specifically on intravitreal injections and anti-VEGF agent use remain scarce, particularly

with respect to drug-specific volumes, prefecture-level crude utilization rates and their inter-prefectural disparities, regional variation in drug choice, and detailed monthly trends.

The COVID-19 pandemic since 2020 has led to reductions in elective surgeries and outpatient visits in many countries [16]. In ophthalmology, temporary declines in cataract and vitreoretinal surgery volumes have been reported [17–20]. For intravitreal injections, reduced visits due to concerns about infection risk and service restrictions at healthcare facilities might theoretically have decreased injection volumes; on the other hand, continued treatment is often necessary to maintain vision. In Japan, the extent to which intravitreal injection volumes changed before, during, and after the COVID-19 pandemic, including possible effects on monthly trends and seasonality, has not been sufficiently clarified.

Therefore, the aim of this study was to describe nationwide patterns of intravitreal injections and anti-VEGF agent use in Japan using NDB Open Data and official population statistics. Specifically, we examined, between fiscal years (FYs) 2017 and 2022: (1) annual trends in the national total number of intravitreal injections and the volumes of each anti-VEGF agent; (2) annual trends and inter-prefectural variation in crude utilization rates (per 100,000 population), with sensitivity analyses accounting for prefecture age structure; (3) regional variation in anti-VEGF drug choice as reflected by prefecture-level shares; (4) monthly changes and seasonality of national injection volumes; and (5) changes in national totals and crude utilization rates before, during, and after the COVID-19 pandemic. Our goal was to provide a nationwide overview of the provision of intravitreal injections and anti-VEGF therapy in Japan and to inform future discussions on resource allocation and service planning.

## Methods

### Study design and data source

This nationwide retrospective observational study was conducted using NDB Open Data, which are aggregated statistics derived from the National Database of Health Insurance Claims and Specific Health Checkups of Japan [21]. The study period covered FYs 2017–2022, with each fiscal year defined as April of one calendar year to March of the following year. Data were obtained from the 4th through 9th releases of NDB Open Data.

From the procedure-specific aggregated tables of NDB Open Data, we identified the procedural code for intravitreal injection (G016, "intravitreal injection") and, using drug-specific codes, the counts of aflibercept, ranibizumab, brolucizumab, and faricimab. In this study, injections of these four agents were collectively defined as intravitreal anti-VEGF injections. The unit of aggregation was the prefecture in which the medical institution was located. For each prefecture and fiscal year, we extracted the number of anti-VEGF injections by care setting (outpatient or inpatient) and by drug.

In the Japanese medical fee schedule, intravitreal injections are reimbursed under procedure code G016. In the NDB Open Data, counts for G016 are aggregated irrespective of the injected drug. Therefore, the total numbers of intravitreal injections used in this study include injections of anti-VEGF agents as well as intravitreal corticosteroids and other intravitreal drugs. In contrast, the use of individual anti-VEGF agents (aflibercept, ranibizumab, brolucizumab, and faricimab) can be identified from drug-specific codes. Across FYs 2017–2022, anti-VEGF agent-specific drug counts accounted for 98.6%–99.3% of all G016 injections (non–anti-VEGF injections, 0.7%–1.4% per year), indicating that G016 serves as a reliable population-level proxy for anti-VEGF injection activity.

Population data were obtained from the official population estimates published by the Statistics Bureau of the Ministry of Internal Affairs and Communications for each fiscal year and prefecture [22]. These data were used to calculate crude utilization rates per 100,000 population, based on the total number of intravitreal injections (outpatient and inpatient combined, including anti-VEGF agents and intravitreal corticosteroids).

### Outcome measures and analytic items

1) **Annual trends in national totals and drug-specific volumes.** For each fiscal year from 2017 to 2022, we calculated

(1) the national total number of intravitreal injections (regardless of drug), and

(2) the national total number of intravitreal anti-VEGF injections and drug-specific volumes for aflibercept, ranibizumab, brolucizumab, and faricimab.

To evaluate annual trends, we constructed simple linear regression models with fiscal year as a continuous independent variable and either the national total number of injections or crude utilization rates as the dependent variable.

To describe changes in national drug shares among anti-VEGF agents, we calculated, for each fiscal year, the proportion of all intravitreal anti-VEGF injections accounted for by each agent and visualized these proportions using bar charts.

**2) Trends and regional variation in crude utilization rates.** Using total intravitreal injection counts (outpatient and inpatient combined) from the NDB Open Data, including injections of both anti-VEGF agents and intravitreal corticosteroids, we calculated crude utilization rates (per 100,000 population) for each of the 47 prefectures. For each year, we summarized the distribution of crude utilization rates using the national mean, standard deviation (SD), minimum, maximum, maximum-to-minimum ratio, and coefficient of variation (CV). Annual changes in the national mean were evaluated using simple linear regression.

Because age structure differs substantially across prefectures and changed over the study period, we performed two sensitivity analyses to assess the impact of population aging. First, we recalculated prefecture-level rates using the population aged ≥75 years as the denominator (injections per 100,000 population aged ≥75 years) for FYs 2017 and 2022, as an index of injection volume relative to the size of the oldest age group. Second, we performed indirect age-standardization based on two age groups (<75 and ≥75 years): expected injection counts for each prefecture were derived by applying national age-specific injection rates to prefecture-specific age-group population counts; standardized ratios and indirectly age-standardized utilization rates were then computed.

***Major metropolitan areas vs other prefectures.*** Major metropolitan areas were defined as Tokyo, Kanagawa, Saitama, Chiba, Aichi, Osaka, Hyogo, and Fukuoka (8 prefectures). The remaining 39 prefectures were classified as non-metropolitan. These definitions were chosen a priori for pragmatic reasons and were not intended to represent formal medical service regions.

***Eastern vs western Japan.*** Eastern Japan comprised 18 prefectures (Hokkaido, Aomori, Iwate, Miyagi, Akita, Yamagata, Fukushima, Ibaraki, Tochigi, Gunma, Saitama, Chiba, Tokyo, Kanagawa, Niigata, Yamanashi, Nagano, and Shizuoka). The remaining 29 prefectures were classified as western Japan. This east–west classification was likewise a pragmatic geographic grouping rather than an official healthcare catchment-area definition.

For each fiscal year, we compared mean crude utilization rates between these regional groups. When distributions were consistent with normality, group differences were assessed using Welch's t-test; otherwise, the Mann–Whitney U test was used. To compare prefecture-level values between FY 2017 and FY 2022, we used paired t-tests and Wilcoxon signed-rank tests.

**3) Regional variation in drug selection (prefecture-level shares).** Using the table of prefecture-level drug shares, we calculated, for each fiscal year, the proportion of intravitreal anti-VEGF injections represented by aflibercept and ranibizumab in each prefecture. We summarized the distribution of these proportions using the median, interquartile range, minimum, maximum, and CV.

Regional variation in drug selection was summarized using prefecture-level shares for each anti-VEGF agent and tabulated for FYs 2017–2022 (Supplementary Table S1).

**4) Monthly volumes and seasonality.** From the "monthly totals" table, we extracted national monthly totals of intravitreal injections for FYs 2019–2022 (April–March for each year). For each fiscal year, we plotted line graphs of monthly injection counts to examine temporal patterns. Using data from all four fiscal years combined, we calculated the mean injection count for each calendar month (April–March) to assess seasonality, with a particular focus on whether volumes tended to increase toward the end of the fiscal year (February–March).

To statistically evaluate monthly differences, we used the Friedman test, treating month (April–March) as the within-subject factor and year as the block. Kendall's W was calculated as a measure of effect size.

**5) Temporal patterns before, during, and after the COVID-19 pandemic.** To assess the impact of the COVID-19 pandemic, we classified the four fiscal years analyzed monthly as follows:

• Pre-COVID period: FY 2019 (April 2019–March 2020).

• COVID period: FYs 2020 and 2021 (April 2020–March 2022).

• Post-COVID period: FY 2022 (April 2022–March 2023).

We adopted this three-period classification as a pragmatic summary of the overall impact of the pandemic, recognizing that infection waves and public health measures varied over time. We applied the Friedman test to monthly national totals from FYs 2019–2022, treating year as the within-subject factor and the 12 months as repeated measures. Pairwise comparisons between FY 2019 and each of FYs 2020, 2021, and 2022 were performed using the Wilcoxon signed-rank test and paired t-tests based on corresponding months.

We further compared the combined total number of injections in April–May 2020—the period of the first state of emergency in Japan—with the corresponding period in 2019 to assess short-term changes during the early phase of the pandemic.

For prefecture-level crude utilization rates, we defined, for each prefecture, the mean of FYs 2017–2019 as the "pre-COVID mean" and the mean of FYs 2020–2022 as the "post-COVID mean". Differences between these two periods were evaluated using paired t-tests and Wilcoxon signed-rank tests.

## Statistical analysis

Continuous variables are presented as mean ± SD or median (interquartile range), as appropriate. Annual trends were evaluated using simple linear regression with fiscal year as a continuous independent variable. Group comparisons were performed using Welch's t-test when assumptions were met; otherwise, the Mann–Whitney U test was used. For repeated measures, paired t-tests or Wilcoxon signed-rank tests were applied, and differences across multiple repeated measures were assessed using the Friedman test with Kendall's W as an effect size. Poisson generalized estimating equation (GEE) models were fitted with an exchangeable working correlation structure and a log(population) offset; robust (sandwich) standard errors were used to account for potential overdispersion. All statistical tests were two-sided, and $p$ values $< 0.05$ were considered statistically significant using JMP version 16 (SAS Institute Inc., Cary, NC, USA). Regression slopes are reported with 95% confidence intervals. For Friedman tests, we report the test statistic ($\chi^2$) with degrees of freedom. P values $< 0.001$ are reported as $p < 0.001$. Because the analyses were primarily descriptive and hypothesis-generating, no formal adjustment was applied for multiple comparisons; p values should be interpreted accordingly.

## Ethics statement

This retrospective study used only aggregated, publicly available data from the National Database of Health Insurance Claims and Specific Health Checkups of Japan (NDB) Open Data and official population estimates from the Statistics Bureau of Japan. The datasets used for this research were accessed on 23 November 2025 (NDB Open Data [21]) and 23 November 2025 (population estimates [22]). No author had access, during or after data collection, to any information that could identify individual participants, and it is not possible to identify specific individuals from the aggregated data provided in these sources. According to national guidelines and our institutional policies, analyses based solely on such publicly available, de-identified data do not require review by an institutional ethics committee or the obtainment of informed consent.

# Results

## 1) Annual trends in national totals and drug-specific volumes

The national total number of intravitreal injections (procedure code G016) increased from 600,068 in FY 2017–671,768 in FY 2018, 742,913 in FY 2019, 789,476 in FY 2020, 853,396 in FY 2021, and 936,715 in FY 2022 (Table 1; Fig 1). Simple linear regression showed that the national total increased by an average of 65,000 injections per year (95% CI, 58,700–71,300; $R^2 = 0.995$, $p < 0.001$). Across FYs 2017–2022, anti-VEGF agent-specific drug counts accounted for 98.6%–99.3% of all G016 injections (non–anti-VEGF injections, 6,871–8,599 per year).

By drug, aflibercept injections increased from 440,961 in FY 2017–662,013 in FY 2022. Ranibizumab injections rose from 150,508–168,038 over the same period, peaking at 183,682 injections in FY 2019. Brolucizumab was introduced in FY 2020, with 19,401 injections in FY 2020, 30,630 in FY 2021, and 40,099 in FY 2022. Faricimab became available in FY 2022, with 59,554 injections in its first year.

**Table 1. National intravitreal injection procedures identified by G016 and agent-specific intravitreal anti-VEGF injection counts by fiscal year.**

| Fiscal year | G016 Total | Aflibercept | Ranibizumab | Brolucizumab | Faricimab | Anti-VEGF Sum | Non-VEGF (%) |
|---|---|---|---|---|---|---|---|
| FY 2017 | 600,068 | 440,961 | 150,508 | | | 591,469 | 1.4 |
| FY 2018 | 671,768 | 498,398 | 164,944 | | | 663,342 | 1.3 |
| FY 2019 | 742,913 | 551,378 | 183,682 | | | 735,060 | 1.1 |
| FY 2020 | 789,476 | 594,690 | 167,874 | 19,401 | | 781,965 | 1.0 |
| FY 2021 | 853,396 | 653,958 | 161,937 | 30,630 | | 846,525 | 0.8 |
| FY 2022 | 936,715 | 662,013 | 168,038 | 40,099 | 59,554 | 929,704 | 0.7 |

Non-VEGF (%) indicates the proportion of G016 intravitreal injection procedures not captured by the included anti-VEGF drug codes and may include intravitreal corticosteroids.

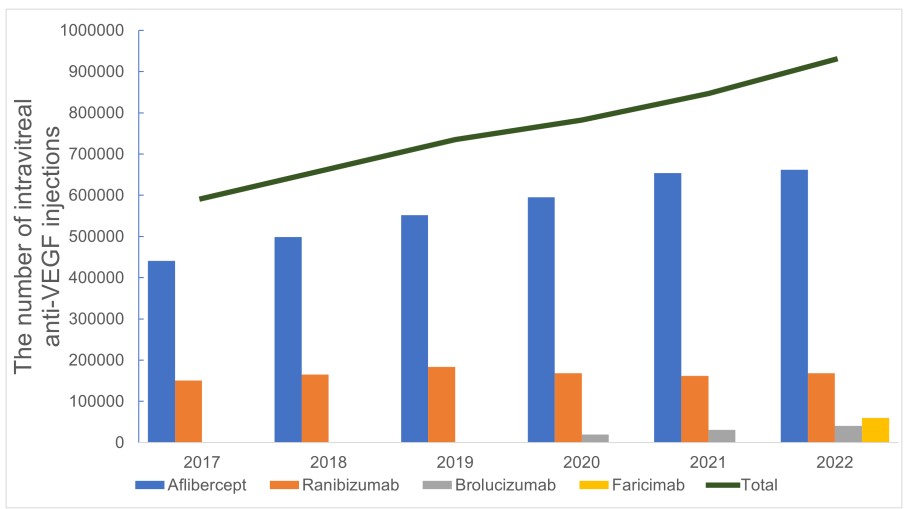

**Fig 1. Annual trends in national intravitreal anti-VEGF injections, fiscal year 2017-2022.** Bars show the annual number of intravitreal injections for each anti-VEGF agent (aflibercept, ranibizumab, brolucizumab, and faricimab), and the solid line indicates the national total number of intravitreal anti-VEGF injections in each fiscal year (FY). Aflibercept and ranibizumab were available throughout the study period, brolucizumab was introduced in FY 2020, and faricimab in FY 2022.

The proportions of each agent among all anti-VEGF injections were as follows: in FY 2017, aflibercept 74.6% and ranibizumab 25.4%; up to FY 2019, these two drugs accounted for almost 100% of anti-VEGF injections. In FY 2020, the shares were aflibercept 76.1%, ranibizumab 21.5%, and brolucizumab 2.5%; in FY 2021, 77.3%, 19.1%, and 3.6%, respectively. In FY 2022, the shares were aflibercept 71.2%, ranibizumab 18.1%, brolucizumab 4.3%, and faricimab 6.4% (Fig 1).

### 2) Trends and regional variation in crude utilization rates

The mean crude utilization rate of intravitreal injections increased from 478.6 ± 109.2 injections per 100,000 population (95% CI of the mean, 446.5–510.7) in FY 2017 to 775.7 ± 143.1 (733.7–817.7) in FY 2022 (Table 2).

Linear regression indicated that the mean rate increased by 57.6 intravitreal injections per 100,000 population per year (95% CI, 52.6–62.5; $R^2 = 0.996$, $p < 0.001$). In FY 2017, prefecture-level crude rates ranged from 199.2 intravitreal injections per 100,000 population in Okinawa to 848.1 in Nagasaki (max/min 4.3; CV 0.228). In FY 2022, the corresponding range was 407.8 (Okinawa) to 1,128.5 (Nagasaki) (max/min 2.8; CV 0.184). In a Poisson generalized estimating equation model with a population offset and repeated measures by prefecture, the estimated annual rate ratio was 1.093 (95% CI 1.085–1.101; $p < 0.001$).

The increase from FY 2017 to FY 2022, expressed as a percentage with FY 2017 set to 100, ranged from 133% to 205% across prefectures (Supplementary Table S2–S4).

When major metropolitan prefectures were compared with non-metropolitan prefectures, mean crude utilization rates were 457.8 ± 72.5 vs 482.8 ± 115.6 injections per 100,000 population in FY 2017 and 726.0 ± 102.3 vs 785.9 ± 149.1 in FY 2022. In a Poisson generalized estimating equation model with a population offset and repeated measures by prefecture, the metropolitan indicator was not associated with the injection rate (IRR 0.94; 95% CI 0.83–1.06; $p = 0.32$), and the time metropolitan interaction was not significant ($p = 0.35$), indicating similar annual growth in both groups.

When eastern Japan was compared with western Japan, mean crude utilization rates were 465.3 ± 100.1 vs 486.8 ± 115.4 injections per 100,000 population in FY 2017 and 745.3 ± 134.3 vs 794.6 ± 147.4 in FY 2022. In a Poisson generalized estimating equation model with a population offset and repeated measures by prefecture, the eastern Japan indicator was not associated with the injection rate (IRR 0.93; 95% CI 0.83–1.04; $p = 0.21$), and the time east interaction was not significant ($p = 0.47$).

Using indirect age standardization based on national age-specific rates (<75 and ≥75 years), variation across prefectures was also attenuated compared with crude utilization rates. In FY 2017, the max/min ratio decreased from 4.3 (crude) to 3.55 (indirect), and the CV decreased from 0.228 to 0.220. In FY 2022, the corresponding values were 2.8 to 2.29 and 0.184 to 0.172, indicating that age structure explains only a modest portion of between-prefecture variation.

**Table 2. Crude utilization rates per 100,000 population across the 47 prefectures by fiscal year.**

| Fiscal year | Mean ± SD | 95% CI (lower) | 95% CI (upper) | Minimum | Maximum | Max/ Min ratio | CV |
|---|---|---|---|---|---|---|---|
| 2017 | 478.6 ± 109.2 | 446.5 | 510.7 | 199.2 | 848.1 | 4.258 | 0.228 |
| 2018 | 540.3 ± 120.6 | 504.9 | 575.8 | 249.2 | 959.5 | 3.851 | 0.223 |
| 2019 | 602.0 ± 125.5 | 565.2 | 638.8 | 306.9 | 1014.7 | 3.306 | 0.208 |
| 2020 | 647.1 ± 130.5 | 608.8 | 685.4 | 322.6 | 1032.1 | 3.199 | 0.202 |
| 2021 | 701.7 ± 131.1 | 663.2 | 740.2 | 369.8 | 1036.5 | 2.803 | 0.187 |
| 2022 | 775.7 ± 143.1 | 733.7 | 817.7 | 407.8 | 1128.5 | 2.767 | 0.184 |

Values are expressed as mean ± standard deviation (SD), 95% confidence interval (CI) of the mean, minimum, maximum, maximum-to-minimum ratio, and coefficient of variation (CV). Rates are based on the total number of intravitreal injections (outpatient and inpatient combined, including anti-VEGF agents and intravitreal corticosteroids).

Observed-to-expected ratios ranged from 0.468 to 1.660 in FY 2017 and from 0.619 to 1.418 in FY 2022 (Supplementary Table S4).

Because prefecture-level injection counts were not available by age group, we conducted sensitivity analyses using the population aged ≥75 years as the denominator, as an index of injection volume relative to the size of the oldest age group. The prefecture-mean rate increased from 3210.6 ± 721.5 to 4681.4 ± 802.1 (injections per 100,000 population aged ≥75 years) from FY 2017 to FY 2022. In FY 2017, rates ranged from 1659.6 to 5176.6 (max/min 3.1; CV 0.225), and in FY 2022 from 2769.6 to 6435.1 (max/min 2.3; CV 0.171). Prefecture rankings were broadly consistent between the total-population and ≥75-year denominators (Spearman's ρ = 0.786 for FY 2017 and 0.768 for FY 2022; both p < 0.001) (Supplementary Tables S2–S3).

### 3) Regional variation in drug selection (prefecture-level shares)

Between FY 2017 and FY 2022, the mean prefecture-level share of aflibercept among anti-VEGF injections decreased slightly from 73% to 70%, with a range of 48–90% in FY 2017 and 55–82% in FY 2022 (mean change −3 percentage points). In contrast, the mean share of ranibizumab decreased more clearly from 21% (range 0–43%) to 14% (0–29%; mean change −7 percentage points). Aflibercept shares increased by ≥10 percentage points in a few prefectures such as Yamagata and Oita, but decreased by ≥10 percentage points in others including Wakayama and Kochi. Ranibizumab shares declined markedly in several prefectures that had high baseline use, such as Yamagata, Yamanashi, Tokushima, Kagawa, Ehime, and Oita, whereas new or increased use of ranibizumab was observed in a small number of prefectures, including Saga and Okinawa. Overall, by FY 2022 aflibercept accounted for around 70% of anti-VEGF injections in most prefectures, while ranibizumab was generally used in 20% or fewer of such injections. Detailed yearly values for each prefecture from FY 2017 to FY 2022 are provided in Supplementary Table S1.

### 4) Monthly volumes and seasonality

National monthly totals for intravitreal injections during FYs 2019–2022 ranged from 59,129–64,273 injections per month in FY 2019, 58,105–74,737 in FY 2020, 66,253–78,890 in FY 2021, and 73,778–85,241 in FY 2022 (Fig 2).

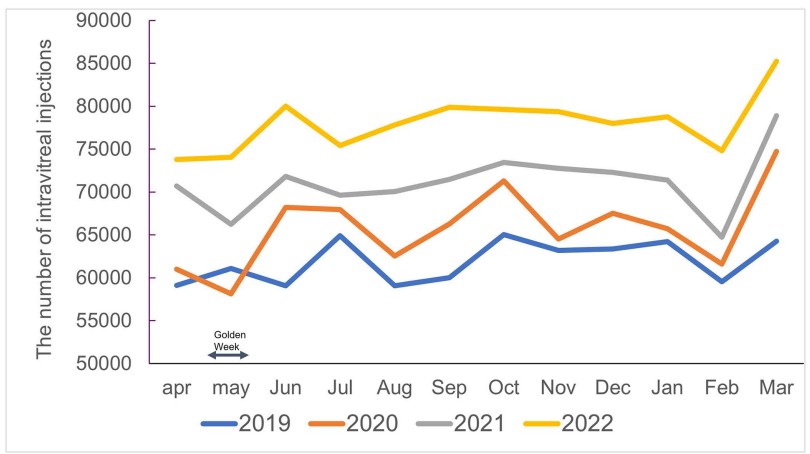

**Fig 2. Monthly number of intravitreal injections from fiscal year 2019 to 2022.** Line graph showing the number of intravitreal injections per month nationwide, from April to the following March, for each fiscal year (2019–2022). Overall monthly counts increased year by year, with broadly similar seasonal patterns and modest peaks toward the end of each fiscal year. The dip in May corresponds to Golden Week, Japan's consecutive national holidays.

Across the four fiscal years, the mean monthly value was highest in March at approximately 76,000 injections and lowest in May at approximately 65,000; the remaining months were generally in the range of 67,000–73,000 injections. The Friedman test using four years of monthly data showed significant differences among months ($\chi^2 = 30.5$, df $= 11$, p $= 0.001$), with Kendall's W $= 0.69$. The mean monthly numbers of injections were 61,909 in FY 2019, 65,790 in FY 2020, 71,116 in FY 2021, and 78,060 in FY 2022, indicating a monotonic increase over time.

**5) Temporal patterns before, during, and after the COVID-19 pandemic**

When monthly totals from FYs 2019–2022 were analyzed as repeated measures, the Friedman test demonstrated significant differences among fiscal years ($\chi^2 = 34.9$, df $= 3$, p $< 0.001$), with a large effect size (Kendall's W $= 0.97$). In Wilcoxon signed-rank tests comparing corresponding months, monthly injection counts in FY 2020, FY 2021, and FY 2022 were all significantly higher than in FY 2019, with median differences of $+3.3 \times 10^3$ injections/month (p $= 0.005$), $+9.2 \times 10^3$ (p $< 0.001$), and $+1.5 \times 10^4$ (p $< 0.001$), respectively.

The mean monthly injection volumes for FY 2019, the combined FYs 2020–2021 (COVID period), and FY 2022 (post-COVID period) were 61,909, 68,453, and 78,060, respectively. Relative to FY 2019 (set as 1.00), the corresponding values were 1.11 and 1.26. The combined total for April–May 2020, during the first state of emergency, was 119,100 injections, representing a 0.9% decrease from the 120,202 injections in April–May 2019.

## Discussion

In this nationwide analysis using NDB Open Data, we characterized annual trends, regional variation, drug selection patterns, monthly changes, and the impact of the COVID-19 pandemic on intravitreal injections in Japan between FYs 2017 and 2022. Total intravitreal injections billed under procedure code G016 increased by approximately 57% (600,068 in FY 2017–936,715 in FY 2022), and agent-specific anti-VEGF injections increased from 591,469–929,704 over the same period. The mean crude utilization rate per 100,000 population across the 47 prefectures increased by about 1.6-fold, and a sensitivity analysis using the population aged ≥75 years as the denominator showed similar temporal increases and a comparable pattern of inter-prefectural variation. Although substantial absolute differences in rates were observed among prefectures, both the coefficient of variation and the maximum-to-minimum ratio decreased slightly, suggesting a modest reduction in relative regional variability. Aflibercept remained the predominant anti-VEGF agent throughout the study period, while the introduction of brolucizumab and faricimab contributed to gradual diversification of drug use. Monthly data showed limited seasonality, with slightly higher volumes toward the end of the fiscal year and lower volumes around major holidays, and national injection volumes were largely maintained during the COVID-19 pandemic.

These findings are broadly consistent with previous Japanese data showing sustained growth in anti-VEGF treatment. Hashimoto and colleagues, using the DeSC claims database, reported that sex- and age-adjusted rates of intravitreal anti-VEGF injections in Japan approximately doubled between 2014 and 2020, that patients aged ≥70 years accounted for about 80% of injections, and that neovascular age-related macular degeneration was the leading indication with aflibercept representing roughly 80% of injections [12]. These observations align with our finding that aflibercept accounted for around 70% of anti-VEGF injections at the national level and that overall intravitreal injection activity continued to expand through FY 2022. Compared with the DeSC database [12], which covers specific insurers and provides detailed patient-level information, NDB Open Data encompass virtually all insured residents and allow population-based prefecture-level comparisons, thereby complementing previous work by extending the observation window, providing national benchmarks, and including newer agents such as faricimab.

Internationally, our results fit into a global pattern of increasing anti-VEGF treatment activity. Large claims-based studies from the United States and Europe have shown steady year-on-year growth in intravitreal anti-VEGF injections, with treatment rates reaching several hundred injections per 100,000 population [13,14,23–25]. At the same time, our data highlight important system-level differences in drug selection. In many Western healthcare systems, bevacizumab

dominates anti-VEGF use because of its low cost and established off-label use, whereas ranibizumab and aflibercept account for smaller proportions. In contrast, bevacizumab is rarely used intravitreally in Japan because it is not approved and not routinely reimbursed for ophthalmic indications, and the market is essentially shared among aflibercept, ranibizumab, brolucizumab, and faricimab [13,14]. Both the present study and previous Japanese claims analyses indicate that aflibercept consistently accounts for the majority of anti-VEGF injections, with ranibizumab generally below 20% by the end of the study period [12]. This dominance of aflibercept likely reflects differences in regulatory approvals, reimbursement policies, pricing structures, and perceived clinical advantages for specific subtypes such as polypoidal choroidal vasculopathy. Faricimab achieved 6.4% market share within its first year of approval (FY 2022), indicating rapid adoption. This may reflect its bispecific mechanism targeting both VEGF-A and angiopoietin-2 (Ang-2) [26,27], and the potential for extended treatment intervals of up to 16 weeks demonstrated in pivotal trials, which align well with treat-and-extend protocols widely used in Japan. In contrast, brolucizumab uptake remained modest (4.3% by FY 2022) despite approval in 2020, likely reflecting clinician caution following post-marketing reports of intraocular inflammation and retinal vasculitis [28,29].

The pronounced inter-prefectural variation in crude utilization rates observed in our study parallels geographic heterogeneity reported in other countries [23]. Although relative variability decreased modestly over time, absolute differences remained large, with FY 2022 rates ranging from approximately 400 to over 1,000 injections per 100,000 population. Importantly, a sensitivity analysis using the population aged ≥75 years as the denominator showed a broadly similar distribution, suggesting that the observed regional differences are not explained solely by variation in the size of older populations. Simple dichotomies such as major metropolitan versus non-metropolitan prefectures or eastern versus western Japan did not yield statistically significant differences in mean rates, suggesting that broad geographic categories alone cannot adequately explain regional variation. More granular factors—such as the distribution of high-volume providers and tertiary centers, referral networks, cross-prefectural patient flow, and regional disease burden—are likely to be more important determinants of access and treatment intensity, but could not be directly assessed using NDB Open Data.

An important methodological feature of this study is that total intravitreal injections were identified using procedure code G016, which does not distinguish between anti-VEGF agents and other intravitreal drugs such as corticosteroids. To address this limitation, we summarized both total G016 procedures and agent-specific anti-VEGF injections identified using drug codes. Across FYs 2017–2022, agent-specific anti-VEGF counts captured 98.6%–99.3% of all G016 procedures, and the residual proportion not captured by the included anti-VEGF codes was small (0.7%–1.4%), which may include intravitreal corticosteroids. These findings support the use of G016 as a reasonable proxy for overall intravitreal injection activity at the population level, while emphasizing that drug-specific analyses should rely on agent codes and that modest regional or temporal differences in non-VEGF use cannot be fully excluded.

Our analysis of monthly injection volumes showed modest but statistically significant differences across months, with lower counts in February and May and higher counts in March. These fluctuations were small compared with the overall upward secular trend, indicating that seasonality in intravitreal injection activity is limited. Factors such as the shorter calendar month and winter conditions in February, which may reduce outpatient visits among elderly patients, and holiday-related scheduling changes during Golden Week in late April to early May may partly explain the lower volumes in these months. Conversely, higher volumes in March may reflect efforts by providers and patients to complete planned treatments before the end of the fiscal year and reimbursement cycle.

With regard to the COVID-19 pandemic, our analysis cannot isolate a causal pandemic effect because we lack a counterfactual scenario (i.e., what would have happened without COVID-19). The observed year-over-year increases primarily reflect the underlying secular trend in intravitreal injection utilization. However, the continuation of this growth trend—rather than a decline or plateau—suggests that intravitreal injection services were maintained as essential care. Whereas Wada et al. reported that cataract extraction declined by 15.5% and vitrectomy by 13.2%

during the first COVID-19 wave (April–May 2020 vs. 2019) using the same NDB Open Data, our data show that intravitreal injection volumes decreased by only 0.9% over the same period. Several international studies have described substantial reductions in anti-VEGF injection visits and treatment delays during the first wave of the pandemic [28,30–34]. In contrast, our national-level analysis showed that monthly intravitreal injection counts in FYs 2020, 2021, and 2022 were all significantly higher than in FY 2019, and even during April–May 2020—the period of the first state of emergency in Japan—the total number of injections was only 0.9% lower than in the corresponding months of 2019. Together with previous NDB-based studies demonstrating sharp but transient declines in cataract and vitreoretinal surgery volumes, these results suggest that intravitreal injections were prioritized and maintained as an essential service. This pattern is consistent with intravitreal injections being relatively preserved as time-sensitive care, given that delaying anti-VEGF therapy may lead to irreversible visual loss. However, the drivers of this difference cannot be determined from aggregated claims data, and temporary delays, regimen modifications, or shifts in drug choice at the individual or facility level cannot be excluded.

From a health-policy perspective, this study underscores the growing and sustained demand for intravitreal injections in an aging society and the need to ensure adequate outpatient capacity and efficient care pathways. The large inter-prefectural differences in crude utilization rates, despite some reduction in relative variability, suggest that access to intravitreal therapy may remain uneven and that structural factors such as travel distance to hospitals and organizational capacity are likely to be relevant. Aligning resource allocation with regional needs, strengthening referral networks, and optimizing injection clinic workflows will be important to sustain high-quality care as the number of eligible patients continues to increase. Future work linking administrative data with disease-specific and facility-level information will be essential to clarify how regional service structures, drug selection patterns, and treatment intensity per patient translate into visual outcomes and long-term healthcare resource use.

This study has several limitations. First, NDB Open Data provide only aggregated counts at the prefectural level; patient-level information, including diagnoses, laterality, and visual outcomes, is not available. As a result, the unit of analysis was the number of injections rather than the number of patients, and we could not distinguish between increases in the number of treated patients and increases in injections per patient. Second, aggregation is based on the location of medical institutions rather than patients' residences, so cross-prefectural referrals and visits to high-volume tertiary centers may have influenced estimates of regional variation. Third, we evaluated crude utilization rates per 100,000 population; because age-specific injection counts by prefecture are not available in NDB Open Data, formal age-standardization across prefectures was not feasible, and we therefore performed a sensitivity analysis using the population aged ≥75 years as the denominator. Fourth, our statistical analyses were primarily descriptive and used simple regression and nonparametric tests; we did not apply formal corrections for multiple comparisons, and prefecture-level inferential results should be interpreted cautiously as exploratory. Finally, the regional groupings used in this study were pragmatic and do not necessarily correspond to formal healthcare catchment areas; more granular spatial analyses will be needed to better understand the determinants of regional variation.

## Conclusion

Using nationwide NDB Open Data, intravitreal injection procedures in Japan increased substantially from FY 2017 to FY 2022. Agent-specific anti-VEGF injections accounted for more than 98% of procedures billed under G016 throughout the study period and increased in parallel with the total G016 volume. Crude utilization rates per 100,000 population rose nationally, and the overall pattern was similar when rates were calculated per 100,000 population aged ≥75 years, while relative inter-prefectural variability narrowed modestly. Aflibercept remained the dominant agent, with gradual diversification after the introduction of brolucizumab and faricimab. Monthly patterns suggested limited seasonality with fiscal year-end and holiday-related fluctuations. National injection volumes were largely preserved during the COVID-19 pandemic, consistent with intravitreal therapy being prioritized as time-sensitive outpatient care.

## Supporting information

**S1 Table. Prefecture-level yearly shares of aflibercept and ranibizumab among intravitreal anti-VEGF injections in Japan, FY2017–FY2022.**
(XLSX)

**S2 Table. Prefecture-level intravitreal injection utilization in FY2017 and FY2022 using crude and ≥75-year population denominators.** Crude utilization rates were calculated as the number of intravitreal injection procedures (G016) per 100,000 total population. Rates using the ≥75-year denominator were calculated as injections per 100,000 population aged ≥75 years, as an index of injection volume relative to the size of the oldest age group. The percentage change represents the relative change in injection counts from FY2017 to FY2022 for each prefecture.
(XLSX)

**S3 Table. Indirect age standardization of intravitreal injection utilization using two age groups (<75 and ≥75 years) in FY2017 and FY2022.** Expected injection counts for each prefecture were derived by applying national age-specific injection rates (<75 and ≥75 years) to prefecture-specific age-group population counts. Observed-to-expected ratios (standardized utilization ratios; observed/expected) with 95% confidence intervals and indirectly age-standardized utilization rates (per 100,000 population) are shown for FY2017 and FY2022.
(XLSX)

**S4 Table. Summary of between-prefecture variation in crude, ≥75-year population–based, and indirectly age-standardized intravitreal injection utilization (FY2017 vs FY2022).** For each fiscal year (FY2017 and FY2022), the distribution of prefecture-level utilization metrics was summarized using the mean, standard deviation (SD), minimum, maximum, maximum-to-minimum ratio, and coefficient of variation (CV = SD/mean) across the 47 prefectures. Metrics include crude utilization rates per 100,000 total population, utilization rates per 100,000 population aged ≥75 years, and indirectly age-standardized utilization rates based on two age groups (<75 and ≥75 years).
(XLSX)

## Acknowledgments

Not applicable.

## Author contributions

**Conceptualization:** Yoshiaki Kabata, Ryo Terauchi.

**Data curation:** Yoshiaki Kabata, Ryo Terauchi.

**Formal analysis:** Yoshiaki Kabata.

**Funding acquisition:** Yoshiaki Kabata.

**Investigation:** Yoshiaki Kabata.

**Methodology:** Yoshiaki Kabata.

**Project administration:** Yoshiaki Kabata.

**Resources:** Yoshiaki Kabata.

**Software:** Yoshiaki Kabata.

**Supervision:** Yoshiaki Kabata.

**Validation:** Yoshiaki Kabata.

**Visualization:** Yoshiaki Kabata.

**Writing – original draft:** Yoshiaki Kabata, Ryo Terauchi.

**Writing – review & editing:** Yoshiaki Kabata, Tadashi Nakano.

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
