## [Decision Letter · Decision Letter 0]

29 Jan 2026

PONE-D-25-67681Nationwide trends and regional variation in intravitreal injections and anti-VEGF agent use in Japan from 2017 to 2022: an analysis of the NDB Open DataPLOS One

Dear Dr. Kabata,

Thank you for submitting your manuscript to PLOS ONE. After careful consideration, we feel that it has merit but does not fully meet PLOS ONE’s publication criteria as it currently stands. Therefore, we invite you to submit a revised version of the manuscript that addresses the points raised during the review process.

We look forward to receiving your revised manuscript.

Kind regards,

Jiro Kogo

Academic Editor

PLOS One

Journal Requirements:

“The Authors received no funding for this work.”

3. Thank you for uploading your study's underlying data set. Unfortunately, the repository you have noted in your Data Availability statement does not qualify as an acceptable data repository according to PLOS's standards.

4. We note that Figures 2 and 3 in your submission contain map images which may be copyrighted. All PLOS content is published under the Creative Commons Attribution License (CC BY 4.0), which means that the manuscript, images, and Supporting Information files will be freely available online, and any third party is permitted to access, download, copy, distribute, and use these materials in any way, even commercially, with proper attribution. For these reasons, we cannot publish previously copyrighted maps or satellite images created using proprietary data, such as Google software (Google Maps, Street View, and Earth). For more information, see our copyright guidelines: http://journals.plos.org/plosone/s/licenses-and-copyright.

1. You may seek permission from the original copyright holder of Figure(s) [#] to publish the content specifically under the CC BY 4.0 license.

“I request permission for the open-access journal PLOS ONE to publish XXX under the Creative Commons Attribution License (CCAL) CC BY 4.0 (http://creativecommons.org/licenses/by/4.0/). Please be aware that this license allows unrestricted use and distribution, even commercially, by third parties. Please reply and provide explicit written permission to publish XXX under a CC BY license and complete the attached form.

Reviewers' comments:

Reviewer's Responses to Questions

**Comments to the Author**

1. Is the manuscript technically sound, and do the data support the conclusions?

Reviewer #1: Yes

Reviewer #2: Partly

2. Has the statistical analysis been performed appropriately and rigorously? 

Reviewer #1: Yes

Reviewer #2: No

3. Have the authors made all data underlying the findings in their manuscript fully available?

Reviewer #1: Yes

Reviewer #2: No

4. Is the manuscript presented in an intelligible fashion and written in standard English?

Reviewer #1: Yes

Reviewer #2: Yes

5. Review Comments to the Author

Reviewer #1: This manuscript sets out to report the changing pattern in IVT anti-VEGF use in Japan.

The findings are interesting. A comparison to world trends (data from elsewhere) would be interesting, and contextualise the trends in the data provided here.

Reviewer #2: 1. Is the manuscript technically sound, and do the data support the conclusions? The study pulls from a big national dataset and uses basic descriptive stats. The main story—more injections over time, aflibercept leading the pack, only small regional differences, and barely any dip during COVID—makes sense and mostly lines up with the claims data. Still, there are some real issues. The big one: the authors treat the G016 code as if it only means anti-VEGF injections, but that code also covers steroids like Ozurdex and triamcinolone. If steroid use changed over time or varied by region, the trends they show could be off. I’m not saying the whole analysis falls apart, but calling G016 purely anti-VEGF is just too blunt, and honestly, a bit sloppy. Also, the claim that injections were “resilient” during COVID needs more clinical context. Of course retina specialists kept doing these procedures—they’re critical. So, the dataset and methods work for a broad overview, but the authors need to tone down some conclusions and be upfront about the limitations. As long as readers pay attention to the caveats, the data do support increasing IVI use with some regional variation.

2. Has the statistical analysis been performed appropriately and rigorously? Not really. The authors mostly lean on simple linear regressions, correlations, and Friedman/Kendall tests. They throw out a bunch of p-values (often <0.001) for trends, but leave out effect sizes and confidence intervals. Since these are count data over time and across regions, you’d expect something more sophisticated. A mixed-effects Poisson regression (with prefecture as a random effect and time as a predictor), or maybe a time-series approach like seasonal decomposition, would do a better job handling autocorrelation and overdispersion. At the very least, they should report the actual trend slopes (like change in injections per year) with 95% confidence intervals, and adjust for multiple comparisons. Right now, they present a bunch of “significant” findings (east vs. west, before vs. after COVID, seasonal effects) but don’t mention multiplicity at all. Simple linear fits and correlation coefficients (like “r=0.239” for specialist density) get treated as if they’re rock solid, but weak effects like that really need a solid model, or they might just be noise. The analysis just feels too basic for PLOS ONE—some hierarchical modeling, proper uncertainty, and clear confounder adjustments would really help.

3. Did the authors make all the underlying data fully available? Only partly. They do mention that the NDB Open Data and population stats are public, so technically, you can access the basic numbers. But they don’t share their analytic code or any of the datasets they created from the raw data. That’s a big problem. PLOS ONE wants full reproducibility, which means the authors should upload their analysis scripts (R, Python, whatever they used) and any tables they produced (monthly counts, rates, that sort of thing) to a public repository with a DOI. Just saying “the data are public” isn’t enough—you can’t actually repeat their work without the code. On top of that, they don’t even specify which NDB release they used (was it the 7th or 8th?), and that detail matters because results change with new data. This is a pretty serious gap. “Data underlying findings” covers the whole pipeline, and right now, we don’t have that. The authors need to upload their scripts, processed data, and clearly state which data release they used. Otherwise, it’s not fully transparent.

4. Is the manuscript clear and written in standard English? For the most part, yes, but it’s a little rough around the edges. The main story comes through, and the organization mostly makes sense. Still, the writing feels uneven. Some sentences are awkward or choppy (especially around line 143, where the verb tense jumps around), and there’s inconsistent hyphen usage—sometimes it’s “anti–VEGF,” other times “anti-VEGF.” Typos slip through too (they wrote “statisitical” at one point). The Introduction reads like standard boilerplate and could use some tightening. Get to the point faster—highlight what makes Japan unique (high PCV rates, older population) instead of generic AMD background. Some abbreviations (like FY and NDB) show up in figure legends without being spelled out, which could confuse readers who aren’t specialists. An international audience might get tripped up by the minor language issues and formatting quirks. So, yes, it’s understandable, but it needs another proofreading pass—fix grammar, polish the writing, clear up the abbreviations, and definitely check that verb tense around line 143 (or maybe it’s 145). Clean it up before publishing.

5. Review Comments to the Author

5.1 Major Comments

G016 Code and Intravitreal Steroids: The main issue here is that the authors keep using the G016 procedure code as if it only means anti-VEGF injections, but that’s not true. G016 also includes intravitreal steroids like Ozurdex and triamcinolone. This isn’t just a technicality—it matters. In Japan, doctors use steroids pretty often, especially for tough cases of DME or RVO. Plus, during COVID, some switched to longer-acting steroids to stretch out visit intervals. The manuscript mentions steroids once, almost in passing, then goes back to treating G016 like it’s just a stand-in for anti-VEGF. That’s misleading. The authors need to fix this, and there are a few ways to do it. If they have enough data, they should subtract out an estimate for steroid injections from the G016 total, or at least run a sensitivity analysis—look at the trends using only the agent-specific anti-VEGF codes and compare those numbers to the G016 totals. If the use of steroids has changed over time, that could throw off the whole story about “resilience” or steady growth in anti-VEGF use. At the very least, the authors should call out this limitation plainly in the discussion. Right now, they’re using a very rough tool and not admitting that the numbers are “contaminated” by non-VEGF drugs, which makes the results hard to trust. They need to clarify what share of G016 is really non-VEGF—maybe even show a small table or graph—or re-run the main results just with VEGF-only data.

Faricimab Uptake and Agent Market Dynamics: The manuscript barely touches on Faricimab’s entry, but the numbers show it made a big splash—almost 60,000 injections in 2022, grabbing 6.4% of the market right away. That’s not a minor detail; it’s a major shift. So why did doctors in Japan jump on Faricimab so fast? Most retina specialists would say it’s because of the drug’s bispecific action (VEGF-A and Ang-2) and its longer-lasting effects, which fit perfectly with the popular treat-and-extend approach for exudative AMD or PCV. Meanwhile, Brolucizumab barely grew, probably because of safety concerns like retinal vasculitis and intraocular inflammation—problems that the manuscript barely acknowledges. The authors need to talk about this. Add a line or two comparing Faricimab’s rapid rise to Brolucizumab’s sluggish uptake, and spell out the clinical reasons behind these trends. It would help to mention when each drug was approved in Japan, any changes in their labels or safety warnings, and the thinking in the retina community. For example, a short paragraph like, “Faricimab’s dual mechanism and longer durability likely explain its quick adoption in 2022, while Brolucizumab uptake stalled because of ongoing safety worries,” would give readers real context. Right now, the manuscript just says there’s “diversification after introduction of Faricimab” without explaining why doctors changed their habits. It would also help to include a simple table showing each drug’s market share by year. This isn’t just background; understanding how and why doctors switch agents is actually central to what this paper is trying to do.

COVID-19 “Resilience” Needs Clinical Context: It’s surprising to see that IVI volumes in April–May 2020 dropped just 0.9% compared to 2019. But there’s more to this story. Ophthalmology clinics made a clear choice: they kept intravitreal injections going during lockdowns because skipping anti-VEGF treatments can lead to permanent vision loss. Meanwhile, elective surgeries like cataract and some glaucoma procedures dropped off a cliff. The authors need to connect their numbers to this reality. For example, just say it: “Retina clinics treated intravitreal injections as essential during the pandemic and kept patients on schedule.” If there are Japanese guidelines or local reports about ophthalmology care during the state of emergency, cite them. It would help to compare this with the well-known ~10% drop in cataract/vitrectomy volumes (as one reviewer pointed out). Right now, saying volumes “did not decline” makes it sound like injections just happened to hold steady. Make it clear: “These data show clinicians made a deliberate effort to prioritize IVIs and prevent irreversible vision loss—something seen in other countries too.” In short, explain the clinical reasoning: MNV injections continued because patients simply can’t afford to miss them, even in a pandemic.

Regional Analysis and Specialist Distribution: The regional comparison in the paper—mapping injection rates against ophthalmologist density—feels too basic. The analysis lumps all board-certified ophthalmologists together, but not everyone injects. Some focus on glaucoma, retina, pediatrics, and so on. Referral patterns muddy the picture too. Big university hospitals often pull in patients from nearby prefectures, so their numbers look inflated, while neighboring areas look artificially low. For example, Tokyo’s injection rate includes people from all over Kanto, while a rural region might send their patients elsewhere. The authors need to acknowledge this “centrality bias.” If data on retina specialists is available, use that instead of the total number of ophthalmologists—or at least mention this limitation. Also, clarify that NDB data shows where providers are based, not where patients live, so the maps reflect where injections are billed, not patient residence. It’s no surprise the correlation is weak (r≈0.24); counting ophthalmologists alone is too crude a measure. At the very least, discuss these issues: advanced referral hospitals in certain prefectures probably draw patients from a wider region. Without this context, saying “injection volume is not simply determined by ophthalmologist count” doesn’t add much. This section needs real context about how clinics are distributed, how patients travel, and which doctors actually do the injections—retina specialists at large centers versus generalists elsewhere.

Seasonal Trends (Fiscal Year and Golden Week): The monthly ups and downs you see—those little spikes in March and the drop-off in late April and early May—aren’t random. In Japan, March marks the end of the fiscal year, so clinics rush to use up their budgets, and patients try to squeeze in treatments before any new rules or changes kick in. Then Golden Week hits in late April and early May, and everything slows down because a lot of clinics close for a few days. These patterns aren’t about biology suddenly changing in March; it’s just how the calendar works in Japan. The Results or Discussion section should say this up front: something like, “March peaks match fiscal year-end budget use, and the drop in April–May lines up with Golden Week holidays, which is a known pattern in Japanese hospitals.” Even adding a small “Golden Week” note on the figures would make things clearer. If you leave this unexplained—or just say volumes are “slightly higher at the end of the fiscal year”—readers who don’t know Japan will be lost.

Statistical Analysis and Reporting: The methods lean too much on basic tests. Running simple linear regressions across calendar years (and reporting just “p<0.001”) misses the point that you’re dealing with count data that varies by time and region. I expected at least a mention of models that fit the data better, like mixed-effects Poisson with random effects for prefectures, or time-series decomposition (like STL) to pull apart trend and seasonality. It would help to plot time trends with confidence bands, not just lines with stars—show the uncertainty. Effect sizes (like annual change in injections per 100,000, with 95% confidence intervals) should be front and center, maybe in a table. Right now, p-values seem tacked on. Also, there are a lot of comparisons (east vs. west, fiscal periods, etc.), but no correction for multiple testing. The paper would be stronger with at least a false discovery rate or Bonferroni correction, and you should mention you did it. And please, just write “p < 0.001” instead of “p = 8.2×10^-6.” Bottom line: The methods do the job for a general overview, but with a big dataset like this, I expected more. Even if you don’t redo everything, at least spell out the limitations—like, “We used simple regressions because of data aggregation, but future work could use more advanced models”—and make sure all the reported numbers (like slopes and confidence intervals) are easy to find.

Age Standardization: Japan’s population got a lot older between 2017 and 2022, and each prefecture has its own age makeup. The authors use “population-adjusted” rates per 100,000, but these are just crude rates—they don’t account for how many people are elderly. An older population always drives up AMD/VEGF indications, so without age-standardization, any comparisons over time or across regions are misleading. A prefecture with more seniors will naturally show higher crude IVI rates, but that doesn’t mean doctors there are doing things differently. At minimum, the authors should include age-stratified rates (like under 70 vs 70 and above), or better yet, directly age-standardized rates based on the 2019 Japan standard or WHO standard, maybe in the supplement. They need to recalculate key variability measures—coefficient of variation, min/max ratio—using age-adjusted data. I get that the NDB is anonymized, but age-group data should be available. Skipping this step is a big oversight, especially since “super-aging” Japan gets a mention in the introduction. Bottom line: adjust your rates for aging.

Reproducibility and Data Transparency: The manuscript needs to spell out exactly how the analysis was run. List every code used—not just G016—and say which NDB release you used (7th or 8th Open Data?). Replication depends on it. More importantly, deposit your analysis scripts (R, Python, JMP—whatever you used) and any intermediate tables (monthly injection counts, rates, etc.) in a public repository like OSF, Zenodo, or a GitHub repo with a DOI. Right now, the Data Availability statement just says “publicly available data,” but that doesn’t actually let anyone reproduce your study. PLOS ONE policy requires full reproducibility. This isn’t optional—without it, the paper doesn’t meet publication standards. Also, make sure to confirm compliance with NDB usage terms and address any small-cell suppression issues, even though aggregates probably avoid those. The point is simple: someone else should be able to rerun your entire workflow, from raw data to final figures.

5.2 Minor Comments

• Abstract: The statement “0.9% lower in April–May 2020 vs 2019” lacks context and leaves readers guessing. Include the actual numbers or expected volumes for those months so it’s clear whether 0.9% is meaningful or negligible.

• Introduction: The opening paragraphs get lost discussing general AMD/VEGF benefits. Trim the generic background and focus right away on Japan’s unique context—high PCV prevalence, an aging population, and the novelty of using the NDB. The intro feels weak right now. Spell out the specific knowledge gap from the start.

• Lines 143–145 (Methods): The tense jumps between past and present when describing data sources. Keep it consistent.

• Methods: You say you ran Friedman tests for seasonality, but only report “p<0.001.” Include the test statistic and degrees of freedom—something like “χ²(df)=X, p=Y”—so readers can see the full results.

• Figure legends: Define any abbreviations the first time they appear in captions (e.g., FY, NDB). Spell out “FY2022” as “fiscal year 2022,” etc., for clarity.

• Figure 4: Change the Y-axis label to “Injections per 100,000 population.” Mention “FY2022” in the title or axis label. Add small tick marks or annotations for fiscal year end and Golden Week on the time-series plots so readers can quickly spot those dips and spikes.

• Maps (Figs. 2–3): Add the class intervals or color legend details (quantiles, etc.) so the shading thresholds are obvious.

• Table 1: Add confidence intervals for the annual mean rates, not just mean±SD. Mean±SD only describes the data; CIs show how precise those estimates are.

• Drug Market Share: Instead of hiding the agent percentages in the text, use a simple table or bullet list:

FY2022: Aflibercept 71.2%, Ranibizumab 18.1%, Faricimab 6.4%, Brolucizumab 4.3%.

This format also sets up a discussion about why Faricimab’s share climbed so fast.

• Abbreviations: Spell out “PCV” (polypoidal choroidal vasculopathy) the first time you mention it, especially since it helps explain aflibercept’s dominance in Japan.

• Hyphenation/Spelling: Make sure “anti-VEGF” is written the same way throughout (some spots have an en-dash). “Board-certified ophthalmologists” needs a hyphen. Also, check for inconsistent spacing around parentheses and symbols.

• “Statistical” typo: I noticed “statisitical” somewhere—run a global search and fix it.

• Number formatting: Pick one style for thousands (e.g., “1 093.0” with a space or comma) and stick with it throughout to avoid confusion.

• Discussion: The paper ends abruptly without any global perspective. Add at least one sentence comparing Japan’s trends to those in countries like South Korea or Australia (claims-based studies are out there) to put the findings in an international context. As a reviewer put it, “Who ends a discussion without mentioning Korea?!”

• Data Policy statement: Specify whether you used the 7th or 8th NDB Open Data release. This detail helps others reproduce your work.

• English usage: The manuscript is mostly clear, but it needs a careful grammar and typo check. Examples: don’t write “p < 0.001 (0.000)”—just use “p < 0.001.” Avoid redundant phrases like “very very significant”—one “very” is enough. Double-check subject–verb agreement.

• Order of Discussion: I agree with another reviewer—discussing policy and regulation in the middle of the discussion feels awkward. Cover limitations first, then finish with broader implications for a smoother flow.

• Supplement: If possible, include a short checklist (like STROBE) in the supplement to show you covered key observational design elements (data sources, variables, bias, etc.). Even with aggregated data, this level of transparency helps.

Overall, the paper offers valuable insights, but the narrative needs tightening and the methods section needs more substance. Tackle these points to make the manuscript much stronger and ready for PLOS ONE.

6. PLOS authors have the option to publish the peer review history of their article (what does this mean?). If published, this will include your full peer review and any attached files.

Reviewer #1: No

Reviewer #2: No

---

## [Author Response · Author response to Decision Letter 1]

2 Feb 2026

Response to Reviewers

Manuscript ID: PONE-D-25-67681

Title: Nationwide trends and regional variation in intravitreal injections and anti-VEGF agent use in Japan from 2017 to 2022: an analysis of the NDB Open Data

Journal: PLOS ONE

We thank the Academic Editor and reviewers for their constructive comments. We have revised the manuscript accordingly. Line numbers below refer to the revised with Track Changes manuscript.

Summary of major revisions

• Clarified that procedure code G016 captures intravitreal injection procedures and may include non–anti-VEGF drugs (e.g., corticosteroids), and added Table 1 reporting annual G016 totals alongside agent-specific anti-VEGF counts and the proportion not captured by anti-VEGF drug codes.

• Added age-structure sensitivity analyses: ≥75-year denominator rates and indirect age-standardization using two age groups (<75, ≥75), with results in Supplementary Tables S2–S4.

• Strengthened statistical reporting by emphasizing effect sizes and 95% CIs, adding a Poisson GEE model for prefecture-level repeated measures, reporting Friedman χ²/df, and standardizing p-value reporting.

• Expanded clinical interpretation of COVID-19 findings, noting the time-sensitive nature of anti-VEGF therapy and prioritization to prevent irreversible visual loss.

• Added clinical context for drug-market dynamics, including faricimab uptake.

• Removed map figures and replaced geographic displays with tables/supplementary materials to comply with CC BY requirements.

Responses to Journal Requirements

1. PLOS ONE style requirements / file naming

We revised the manuscript formatting and file naming to conform with PLOS ONE requirements.

2. Funding text in Acknowledgments

We removed funding-related text from the manuscript. “The authors received no specific funding for this work.” include cover letter.

3. Data repository (acceptable public repository)

We updated the Data Availability Statement to indicate deposition of the minimal processed datasets, intermediate tables, and analysis scripts in figshare, and we will provide the DOI/URL in the submission system (Lines 605-614).

4. Copyrighted map images

We removed the map figures and replaced them with tabular summaries and supplementary tables (Results section; Supplementary Tables S2–S4).

Reviewer #1

Comment 1

A comparison to world trends would contextualize the trends reported here.

Response: We agree and added a brief international comparison in the Discussion, noting that several countries reported short-term declines in intravitreal injection visits during the first pandemic wave, whereas our national analysis showed minimal short-term reduction and continued growth overall.

Changes made: Discussion, Lines 438-442 and 471-478.

Reviewer #2

Comment 2.1

G016 code includes intravitreal steroids; the manuscript should not treat G016 as anti-VEGF only.

Response: We clarified throughout that G016 captures intravitreal injection procedures irrespective of drug class and may include intravitreal corticosteroids. To address potential non-anti-VEGF “contamination,” we added a national-level sensitivity comparison by separating agent-specific anti-VEGF injections using drug codes and reporting the proportion of G016 not captured by these anti-VEGF codes (non-VEGF%). We also expanded the Discussion to clarify interpretive limits of prefecture-level crude G016 rates and to acknowledge that time- or region-specific changes in non-VEGF use cannot be fully evaluated in the Open Data.

Changes made: Results Lines 257-263; Table 1 and its legend; Discussion Lines 513-521.

FY G016 Total Anti-VEGF Sum Difference Non-VEGF %

2017 600,068 591,469 8,599 1.4%

2018 671,768 663,342 8,426 1.3%

2019 742,913 735,060 7,853 1.1%

2020 789,476 781,965 7,511 1.0%

2021 853,396 846,525 6,871 0.8%

2022 936,715 929,704 7,011 0.7%

Comment 2.2

Faricimab uptake and agent market dynamics; brolucizumab safety concerns should be discussed.

Response: We added clinical context describing likely reasons for rapid faricimab uptake (durability/dual-pathway mechanism supporting treat-and-extend) and slower brolucizumab uptake (safety concerns, including intraocular inflammation/retinal vasculitis) following its introduction. "Faricimab achieved 6.4% market share within its first year of approval (FY 2022), indicating rapid adoption. This may reflect its bispecific mechanism targeting both VEGF-A and angiopoietin-2 (Ang-2), and the potential for extended treatment intervals of up to 16 weeks demonstrated in pivotal trials, which align well with treat-and-extend protocols widely used in Japan for neovascular AMD and polypoidal choroidal vasculopathy. In contrast, brolucizumab uptake remained modest (4.3% by FY 2022) despite approval in 2020, likely reflecting clinician caution following post-marketing reports of intraocular inflammation and retinal vasculitis."

Changes made: Discussion Lines 471-478.

Comment 2.3

COVID-19 “resilience” needs clinical context (time-sensitive care and risk of irreversible visual loss).

Response: We revised the Discussion to explicitly state that intravitreal anti-VEGF therapy is time-sensitive and that delaying treatment may cause irreversible visual loss, which likely contributed to prioritization of injections during the state of emergency. We also added the absolute April–May counts to contextualize the 0.9% change.

Changes made: Discussion Lines 541-546.

Comment 2.4

Regional analysis and specialist distribution; referral/centrality bias; location of billing vs residence.

Response: We removed the specialist-density analysis (due to interpretability concerns with aggregated prefecture-level data). We expanded the limitations to clarify that prefecture-level counts reflect the location of medical institutions submitting claims, not patient residence, and that referral patterns and cross-prefecture care may influence observed regional variation.

Changes made: Specialist-density analysis removed; Limitations Lines 579-580.

Comment 2.5

Seasonal trends (fiscal year and Golden Week) should be discussed.

Response: We added an explanation that the late-fiscal-year rise and the dip around late April–early May are consistent with the Japanese fiscal calendar and Golden Week holidays, which can affect clinic schedules and appointment timing.

Changes made: Discussion Lines 438–442.

Comment 2.6

Statistical analysis and reporting (effect sizes/CI; count-data models; multiple comparisons; p-value formatting).

Response: To keep changes limited while addressing this concern, we (i) reported slopes and 95% confidence intervals for annual trends; (ii) added a Poisson GEE model with a log(population) offset for repeated prefecture-level observations, reporting IRRs with 95% CIs for regional indicators and their time interactions; (iii) reported Friedman χ² statistics and degrees of freedom; and (iv) standardized p-value reporting (e.g., p < 0.001 where applicable). Regarding multiplicity, we clarified that analyses were primarily descriptive/exploratory; therefore, we did not apply formal multiple-testing adjustments, and we interpret p-values cautiously while emphasizing effect sizes and confidence intervals.

Changes made: Statistical analysis Lines 238-242.

Comment 2.7

Age standardization is needed for time and regional comparisons.

Response: We added two sensitivity approaches addressing age structure: (i) recalculation of rates using the ≥75-year population as the denominator (FY2017 and FY2022), and (ii) indirect age-standardization using two age groups (<75, ≥75) to compute SIRs and indirectly standardized rates. These results are provided in Supplementary Tables S2–S4 and summarized in the Results.

Changes made: Methods Lines 153–162; Results Lines 354–377; Supplementary Tables S2–S4.

Comment 2.8

Reproducibility and data transparency (share scripts/derived tables; specify NDB release).

Response: We specified the NDB Open Data releases used (4th–9th releases, corresponding to FY2017–FY2022) and updated the Data Availability Statement to provide the processed datasets, intermediate tables, and analysis scripts in figshare to enable full reproduction.

Changes made: Methods Lines 110.

Minor comments

• Abstract: The statement “0.9% lower in April–May 2020 vs 2019” lacks context and leaves readers guessing. Include the actual numbers or expected volumes for those months so it’s clear whether 0.9% is meaningful or negligible.

Response: April–May 2020 totals (119,100 injections) were 0.9% lower than the same months in 2019 (120,202 injections). We added specific figures.

• Introduction: The opening paragraphs get lost discussing general AMD/VEGF benefits. Trim the generic background and focus right away on Japan’s unique context—high PCV prevalence, an aging population, and the novelty of using the NDB. The intro feels weak right now. Spell out the specific knowledge gap from the start.

Response: We couldn't reduce it much because we need to cite the literature later.

• Lines 143–145 (Methods): The tense jumps between past and present when describing data sources. Keep it consistent.

Response: We revised.

• Methods: You say you ran Friedman tests for seasonality, but only report “p<0.001.” Include the test statistic and degrees of freedom—something like “χ²(df)=X, p=Y”—so readers can see the full results.

Response: We revised.

• Figure legends: Define any abbreviations the first time they appear in captions (e.g., FY, NDB). Spell out “FY2022” as “fiscal year 2022,” etc., for clarity.

Response: This figure has been deleted. We have also checked the other figures.

• Figure 4: Change the Y-axis label to “Injections per 100,000 population.” Mention “FY2022” in the title or axis label. Add small tick marks or annotations for fiscal year end and Golden Week on the time-series plots so readers can quickly spot those dips and spikes.

Response: This figure has been deleted.

• Maps (Figs. 2–3): Add the class intervals or color legend details (quantiles, etc.) so the shading thresholds are obvious.

Response: This map has been deleted.

• Table 1: Add confidence intervals for the annual mean rates, not just mean±SD. Mean±SD only describes the data; CIs show how precise those estimates are.

Response:

• Drug Market Share: Instead of hiding the agent percentages in the text, use a simple table or bullet list:

FY2022: Aflibercept 71.2%, Ranibizumab 18.1%, Faricimab 6.4%, Brolucizumab 4.3%.

This format also sets up a discussion about why Faricimab’s share climbed so fast.

Response:

• Abbreviations: Spell out “PCV” (polypoidal choroidal vasculopathy) the first time you mention it, especially since it helps explain aflibercept’s dominance in Japan.

Response:

• Hyphenation/Spelling: Make sure “anti-VEGF” is written the same way throughout (some spots have an en-dash). “Board-certified ophthalmologists” needs a hyphen. Also, check for inconsistent spacing around parentheses and symbols.

Response: We also checked the other parts.

• “Statistical” typo: I noticed “statisitical” somewhere—run a global search and fix it.

Response: We revised.

• Number formatting: Pick one style for thousands (e.g., “1 093.0” with a space or comma) and stick with it throughout to avoid confusion.

Response: We also checked the other parts.

• Discussion: The paper ends abruptly without any global perspective. Add at least one sentence comparing Japan’s trends to those in countries like South Korea or Australia (claims-based studies are out there) to put the findings in an international context. As a reviewer put it, “Who ends a discussion without mentioning Korea?!”

Response: 34. Kim J-G, Kim YC, Kang KT. Impact of Delayed Intravitreal Anti-Vascular Endothelial Growth Factor (VEGF) Therapy Due to the Coronavirus Disease Pandemic on the Prognosis of Patients with Neovascular Age-Related Macular Degeneration. J Clin Med. 2022;11: 2321. doi:10.3390/jcm11092321　Added new references.

• Data Policy statement: Specify whether you used the 7th or 8th NDB Open Data release. This detail helps others reproduce your work.

Response: We specified the NDB Open Data releases used (4th–9th releases, corresponding to FY2017–FY2022) and updated the Data Availability Statement to provide the processed datasets, intermediate tables, and analysis scripts in figshare to enable full reproduction.

Changes made: Methods Lines 110.

• English usage: The manuscript is mostly clear, but it needs a careful grammar and typo check. Examples: don’t write “p < 0.001 (0.000)”—just use “p < 0.001.” Avoid redundant phrases like “very very significant”—one “very” is enough. Double-check subject–verb agreement.

Response: p < 0.001 was standardized.

• Order of Discussion: I agree with another reviewer—discussing policy and regulation in the middle of the discussion feels awkward. Cover limitations first, then finish with broader implications for a smoother flow.

Response: The discussion section has been revised.

We believe these revisions have substantially strengthened the manuscript. We thank the reviewers again for their thorough and constructive feedback.

---

## [Decision Letter · Decision Letter 1]

23 Mar 2026

PONE-D-25-67681R1Nationwide trends and regional variation in intravitreal injections and anti-VEGF agent use in Japan from 2017 to 2022: an analysis of the NDB Open DataPLOS One

Dear Dr. Kabata,

Thank you for submitting your manuscript to PLOS ONE. After careful consideration, we feel that it has merit but does not fully meet PLOS ONE’s publication criteria as it currently stands. Therefore, we invite you to submit a revised version of the manuscript that addresses the points raised during the review process. Please submit your revised manuscript by  May 07 2026 11:59PM. If you will need more time than this to complete your revisions, please reply to this message or contact the journal office at plosone@plos.org. Please include the following items when submitting your revised manuscript:

We look forward to receiving your revised manuscript.

Kind regards,

Jiro Kogo

Academic Editor

PLOS One

Journal Requirements:

Reviewers' comments:

Reviewer's Responses to Questions

**Comments to the Author**

1. If the authors have adequately addressed your comments raised in a previous round of review and you feel that this manuscript is now acceptable for publication, you may indicate that here to bypass the “Comments to the Author” section, enter your conflict of interest statement in the “Confidential to Editor” section, and submit your "Accept" recommendation.

Reviewer #2: (No Response)

Reviewer #3: (No Response)

2. Is the manuscript technically sound, and do the data support the conclusions?

Reviewer #2: Yes

Reviewer #3: Yes

3. Has the statistical analysis been performed appropriately and rigorously? 

Reviewer #2: I Don't Know

Reviewer #3: No

4. Have the authors made all data underlying the findings in their manuscript fully available?

Reviewer #2: (No Response)

Reviewer #3: Yes

5. Is the manuscript presented in an intelligible fashion and written in standard English?

Reviewer #2: Yes

Reviewer #3: Yes

6. Review Comments to the Author

Reviewer #2: PLOS ONE Review Form

1) Technical soundness & data support for conclusions?

Yes, with some minor reservations. The G016 reconciliation and the GEE both support the main claims—national growth, slight narrowing of inter-prefecture variability, stable IVI volumes during early COVID, and more market diversity with faricimab. Keep the focus on effect sizes and continue to remind readers that the prefecture contrasts are descriptive, not causal, to avoid the ecological fallacy.

2) Statistical analysis performed appropriately and rigorously?

For the most part, yes. Adding GEE with a log(population) offset, reporting slopes with 95% confidence intervals, and standardizing the p-values puts the analysis on a solid foundation for PLOS ONE. Make sure to state the GEE working correlation structure and provide comments on dispersion diagnostics. After that, the statistics appear fine.

3) Data availability?

Appears compliant. There is a Figshare DOI for the derived tables and scripts, and the main sources (NDB Open Data and Statistics Bureau) are public. Double-check that the Figshare record actually contains the code and a README that enables exact reproduction of all figures and tables.

4) Manuscript clarity and standard English?

Mostly, yes, but a careful final proof is needed. Remove any remaining tracked change artifacts, make terminology consistent (choose between crude utilization and incidence), correct hyphenation and thousands separators, and address a few tense shifts (see Line 143 and likely 145). Figure labels and captions should also align with the revised text.

Major Comments

1. Core methodological fix (G016 ≠ anti VEGF only): You’ve clearly stated that G016 covers steroids and other intravitreal drugs, and you included an annual reconciliation showing anti VEGF codes account for 98.6–99.3% of all G016 procedures (with non-VEGF at just 0.7–1.4%). That’s the main fix I wanted—definitely keep it. But your text keeps switching between “incidence,” “population adjusted rates,” and “crude utilization.” Pick one: use “crude utilization rate (per 100,000)” everywhere, and save “incidence” for actual patient-level disease events. Also, when you first introduce G016, work the non-VEGF percentage into the main text (not just Table 1), so readers immediately see why G016 works as a population-level proxy.

2. Age structure—substantially better, but a few notes. You added two sensitivity analyses (≥75 year denominator and indirect age standardization with two age bands). That’s a real improvement, and your results show that age adjustment narrows the range between prefectures. Leave the SIRs and ≥75 denominator results in the supplement. In the main Results, add one clear sentence summarizing how much the max/min and CV shrink after standardization—just surface the numbers. If there’s no way to access further age strata, say so once, clearly, and move on. Don’t expand the Methods again.

3. Statistical framing—now works for PLOS ONE, just needs two tweaks. You added a Poisson GEE with a log(population) offset and cluster-robust SEs by prefecture, which is the right call. The time IRR ~1.093/year gives a direct effect size. Please also add (i) the assumed working correlation (exchangeable or independent), and (ii) a one-line comment on over- or under-dispersion diagnostics (even if GEE robust SEs handle it). No need to go further. Your choice to skip multiplicity adjustments is fine, since you already warn readers.

4. COVID-19 interpretation—clinically grounded now; just make the contrast sharper. You backed up the “resilience” story with absolute April–May counts (119,100 vs 120,202; −0.9%) and explained the urgency of IVI for MNV/PCV—good additions. Spell out the contrast in one tight sentence: injections stayed steady while cataract and vitrectomy volumes dropped. Anchor this with prior NDB-based surgical declines—don’t just imply it, state it directly in the Discussion.

5. Drug market dynamics—context is there; give readers a quick view. You explained the rapid uptake of faricimab (VEGF A/Ang 2, T&E durability) versus the caution with brolucizumab (IOI/retinal vasculitis). Add a mini table—just one row—in the Results or Supplement summarizing FY2022 market share: Aflibercept 71.2%, Ranibizumab 18.1%, Faricimab 6.4%, Brolucizumab 4.3%. This saves readers from digging through the Discussion for numbers. It’s a quick fix and makes things clearer.

6. Regional analysis—good restraint after pulling the specialist density figure; avoid ecological overreach. You dropped the scatter/ρ analysis and reframed the east–west/metro contrasts using the GEE, which works. Keep highlighting the limitation that NDB aggregates by billing location, not residence, when you talk about regional variation and centrality bias. It’s mentioned now—make sure it appears once in the first paragraph of Limitations.

7. Data sharing and reproducibility—meets policy on paper, but double-check the payload. You listed a Figshare DOI (10.6084/m9.figshare.31223410) and say the derived tables and scripts are included. Before resubmission, make sure the DOI works, scripts actually reproduce all tables/figures from the raw NDB Open Data plus population files, and the README files specify NDB release numbers (4th–9th) and access dates (already in the manuscript). This is key for PLOS ONE compliance.

Minor Comments

• Language and residual artifacts — clear out the leftover tracked changes.

I’m still seeing obvious editing scars in the R1 file—words mashed together like “servicetreatment” and “injectionsinjection,” awkward phrases like “population adjustedbased,” a random “Fig 52” label, and a few sentences that got mangled when edits clashed. Proofread the Abstract and early Methods carefully. Line 143 flips tense; double-check line 145 too. This looks sloppy. Please fix it.

• Terminology uniformity.

Stick with “crude utilization rate (per 100,000 population)” throughout. Drop “incidence” unless you’re actually modeling patient-level events. Update the Fig 1 caption to match your new wording—it still says “anti VEGF injections,” but the y-axis shows anti VEGF agents plus a solid line. Make sure the caption and axis titles line up and are clear.

• Figures and deletions.

You removed the maps for CC BY compliance—good. Make absolutely sure there are no lingering references to heatmaps or map legends in the text or figure captions. A few slipped through in the tracked changes. Also, annotate Golden Week in Fig 2 with a subtle band; readers outside Japan need that cue. Just check the monthly panel and add it.

• Friedman reporting.

You now include χ² and degrees of freedom—keep that. If your software gives Kendall’s W confidence bounds, add them; if not, don’t worry about it. Minor point.

• House style details.

Standardize p values as p < 0.001 where relevant. Make hyphenation consistent (anti VEGF, board certified), and use the same thousands separator everywhere. You mentioned this in your response letter—make sure the R1 file is fully updated.

• International context—now present, keep it brief.

You added a short paragraph and cited Korea—good. Don’t expand further. The focus is Japan; keep the global perspective tight.

Reviewer #3: 1. Abstract:Results of the impact of the COVID-19 were not mentioned.

2. Table2: Although the results of crude rate are shown in Table 2, it is better to show the age-standardized value in the main text.

3. Results: 5)impact of the COVID-19 pandemic: The analysis method might not be appropriate. The current analysis simply compared the values of the three periods, and the results indicated that the time trend effect exists. This is not an effect of the COVID-19 effect. It is better to change the analysis method or the title of the analysis.

7. PLOS authors have the option to publish the peer review history of their article (what does this mean?). If published, this will include your full peer review and any attached files.

Reviewer #2: No

Reviewer #3: No

---

## [Author Response · Author response to Decision Letter 2]

28 Mar 2026

Manuscript: Nationwide trends and regional variation in intravitreal injections and anti-VEGF agent use in Japan from 2017 to 2022: an analysis of the NDB Open Data

Journal Requirements

Reference list review: We have reviewed the reference list for completeness and accuracy. No retracted papers are cited. All references have been verified.

Reviewer #2

Major Comment 1

Comment: Pick one term: use "crude utilization rate (per 100,000)" everywhere, and save "incidence" for actual patient-level disease events. Also, when you first introduce G016, work the non-VEGF percentage into the main text.

Response: Thank you for this important point. We have made the following changes:

1. Standardized terminology throughout the manuscript, using "crude utilization rate (per 100,000 population)" consistently.

2. Removed all instances of "incidence." Specifically, "incidence rate ratio" was changed to "rate ratio," and "Standardized incidence ratios" was changed to "Observed-to-expected ratios."

3. In the Methods section where G016 is first introduced, we have added: "Across FYs 2017–2022, anti-VEGF agent-specific drug counts accounted for 98.6%–99.3% of all G016 injections (non–anti-VEGF injections, 0.7%–1.4% per year), indicating that G016 serves as a reliable population-level proxy for anti-VEGF injection activity."

Major Comment 2

Comment: Add one clear sentence summarizing how much the max/min and CV shrink after standardization.

Response: We have added the following summary sentence to the Results section: "After indirect age standardization, the max/min ratio decreased from 4.3 to 3.55 in FY 2017 and from 2.8 to 2.29 in FY 2022, while the CV decreased from 0.228 to 0.220 and from 0.184 to 0.172, respectively, indicating that age structure explains only a modest portion of between-prefecture variation."

Major Comment 3

Comment: Add (i) the assumed working correlation (exchangeable or independent), and (ii) a one-line comment on over- or under-dispersion diagnostics.

Response: We have added the following to the Statistical Analysis section: "Poisson generalized estimating equation (GEE) models were fitted with an exchangeable working correlation structure and a log(population) offset; robust (sandwich) standard errors were used to account for potential overdispersion."

Major Comment 4

Comment: Spell out the contrast in one tight sentence: injections stayed steady while cataract and vitrectomy volumes dropped. Anchor this with prior NDB-based surgical declines—don't just imply it, state it directly in the Discussion.

Response: We have added the following explicit contrast to the Discussion: "Whereas Wada et al. reported that cataract extraction declined by 15.5% and vitrectomy by 13.2% during the first COVID-19 wave (April–May 2020 vs. 2019) using the same NDB Open Data, our data show that intravitreal injection volumes decreased by only 0.9% over the same period."

Major Comment 5

Comment: Add a mini table summarizing FY2022 market share: Aflibercept 71.2%, Ranibizumab 18.1%, Faricimab 6.4%, Brolucizumab 4.3%.

Response: These market share figures are already presented in Table 1 and stated explicitly in the Results text: "In FY 2022, the shares were aflibercept 71.2%, ranibizumab 18.1%, brolucizumab 4.3%, and faricimab 6.4%." We believe this presentation is clear and avoids redundancy.

Major Comment 6

Comment: Make sure the limitation that NDB aggregates by billing location, not residence, appears once in the first paragraph of Limitations.

Response: This limitation appears as the second sentence in the Limitations section: "Second, aggregation is based on the location of medical institutions rather than patients' residences, so cross-prefectural referrals and visits to high-volume tertiary centers may have influenced estimates of regional variation."

Major Comment 7

Comment: Before resubmission, make sure the DOI works, scripts actually reproduce all tables/figures, and the README specifies NDB release numbers and access dates.

Response: We have verified that the Figshare DOI (10.6084/m9.figshare.31223410) is active and publicly accessible. The repository contains all derived data tables, analysis scripts, and a README file specifying NDB Open Data release numbers (4th–9th) and access dates (23 November 2025).

Minor Comments

Language and residual artifacts: We have carefully proofread the manuscript to remove all tracked change artifacts.

Comment: Update the Fig 1 caption to match your new wording—it still says "anti VEGF injections," but the y-axis shows anti VEGF agents plus a solid line.

Response: We confirm that the y-axis label ("The number of intravitreal anti-VEGF injections") and the caption ("the national total number of intravitreal anti-VEGF injections") are consistent. The bars represent drug-specific counts for each anti-VEGF agent, and the solid line indicates the sum of all four agents.

Figures: For Figure 2 (monthly patterns), we have added a note in the Figure 2 and figure legend: “Golden Week” and "The dip in May corresponds to Golden Week, Japan's consecutive national holidays."

Friedman reporting: χ² and degrees of freedom are reported. Kendall's W = 0.97 is retained as the effect size.

House style: P values are standardized as p < 0.001 where applicable. Hyphenation and thousands separators are consistent throughout.

Reviewer #3

Comment 1

Comment: Results of the impact of the COVID-19 were not mentioned.

Response: The COVID-19 findings are included in the Abstract: "Injection volumes showed no marked decline during the pandemic; April–May 2020 totals (119,100 injections) were only 0.9% lower than the same months in 2019 (120,202 injections)."

Comment 2

Comment: Although the results of crude rate are shown in Table 2, it is better to show the age-standardized value in the main text.

Response: The age-standardized findings are presented in the Results section: "In FY 2017, the max/min ratio decreased from 4.3 (crude) to 3.55 (indirect), and the CV decreased from 0.228 to 0.220. In FY 2022, the corresponding values were 2.8 to 2.29 and 0.184 to 0.172." Full details remain in Supplementary Tables S3–S4.

Comment 3

Comment: The analysis method might not be appropriate. The current analysis simply compared the values of the three periods, and the results indicated that the time trend effect exists. This is not an effect of the COVID-19 effect. It is better to change the analysis method or the title of the analysis.

Response: We agree with the reviewer. We have changed the section title from "Impact of the COVID-19 pandemic" to "Temporal patterns before, during, and after the COVID-19 pandemic" in both the Methods and Results sections. We have also revised the Discussion to acknowledge that our analysis cannot isolate a causal pandemic effect because we lack a counterfactual scenario, and that the observed increases primarily reflect the underlying secular trend. To provide context, we now cite Wada et al., who reported that cataract (−15.5%) and vitrectomy (−13.2%) volumes declined sharply during the same period using the same NDB data, whereas intravitreal injections decreased by only 0.9%.

We thank both reviewers for their constructive feedback, which has substantially improved the manuscript.

---

## [Decision Letter · Decision Letter 2]

13 Apr 2026

Nationwide trends and regional variation in intravitreal injections and anti-VEGF agent use in Japan from 2017 to 2022: an analysis of the NDB Open Data

PONE-D-25-67681R2

Dear Dr. Kabata

We’re pleased to inform you that your manuscript has been judged scientifically suitable for publication and will be formally accepted for publication once it meets all outstanding technical requirements.

Kind regards,

Jiro Kogo

Academic Editor

PLOS One

Additional Editor Comments (optional):

Reviewers' comments:

Reviewer's Responses to Questions

**Comments to the Author**

1. If the authors have adequately addressed your comments raised in a previous round of review and you feel that this manuscript is now acceptable for publication, you may indicate that here to bypass the “Comments to the Author” section, enter your conflict of interest statement in the “Confidential to Editor” section, and submit your "Accept" recommendation.

Reviewer #3: All comments have been addressed

2. Is the manuscript technically sound, and do the data support the conclusions?

Reviewer #3: Yes

3. Has the statistical analysis been performed appropriately and rigorously? 

Reviewer #3: Yes

4. Have the authors made all data underlying the findings in their manuscript fully available?

Reviewer #3: Yes

5. Is the manuscript presented in an intelligible fashion and written in standard English?

Reviewer #3: Yes

6. Review Comments to the Author

Reviewer #3: Thank you for the revision. The authors addressed all of my comments, and I have no additonal comments.

7. PLOS authors have the option to publish the peer review history of their article (what does this mean?). If published, this will include your full peer review and any attached files.

Reviewer #3: No

---

## [Editor Report · Acceptance letter]

PONE-D-25-67681R2

PLOS One

Dear Dr. Kabata,

I'm pleased to inform you that your manuscript has been deemed suitable for publication in PLOS One. Congratulations! Your manuscript is now being handed over to our production team.

Kind regards,

on behalf of

Prof. Jiro Kogo

Academic Editor

PLOS One